# Nano Ranking Analysis: determining NPF event occurrence and intensity based on the concentration spectrum of formed (sub-5 nm) particles

Diego Aliaga[1], Santeri Tuovinen[1], Tinghan Zhang[1], Janne Lampilahti[1], Xinyang Li[1], Lauri Ahonen[1], Tom Kokkonen[1], Tuomo Nieminen[1,2], Simo Hakala[1], Pauli Paasonen[1], Federico Bianchi[1], Doug Worsnop[1,3], Veli-Matti Kerminen[1], Markku Kulmala[1,4,5]

[1]Institute for Atmospheric and Earth System Research (INAR) / Physics, Faculty of Science, University of Helsinki, Finland
[2]Department of Physics, Faculty of Science, University of Helsinki, Finland
[3]Aerodyne Research Inc., Billerica, Massachusetts, United States
[4]Aerosol and Haze Laboratory, Beijing Advanced Innovation Center for Soft Matter Sciences and Engineering, Beijing University of Chemical Technology (BUCT), Beijing, China
[5]Joint International Research Laboratory of Atmospheric and Earth System Sciences, School of Atmospheric Sciences, Nanjing University, Nanjing, China

*Correspondence to*: Markku Kulmala (markku.kulmala@helsinki.fi)

**Abstract.** Here we introduce a new method, termed "Nano Ranking Analysis," for characterizing new particle formation (NPF) from atmospheric observations. Using daily variations of the particle number concentration at sizes immediately above the continuous mode of molecular clusters, here in practice 2.5-5 nm - $\Delta N_{2.5-5}$, we can determine the occurrence probability and estimate the strength of atmospheric NPF events. After determining the value of $\Delta N_{2.5-5}$ for all the days during a period under consideration, the next step of the analysis is to rank the days based on this simple metric. The analysis is completed by grouping the days either into a number of percentile intervals based on their ranking or into a few modes in the distribution of $\log(\Delta N_{2.5-5})$ values. Using five years (2018-2022) of data from the SMEAR II station in Hyytiälä, Finland, we found that the days with higher (lower) ranking values had, on average, both higher (lower) probability of NPF events and higher (lower) particle formation rates. The new method provides probabilistic information about the occurrence and intensity of NPF events and is expected to serve as a valuable tool to define the origin of newly formed particles at many types of environments that are affected by multiple sources of aerosol precursors.

## 1 Introduction

Atmospheric new particle formation (NPF) events take place, though with variable frequencies, in most of the continental environments (e.g Wang et al., 2017; Kerminen et al., 2018; Nieminen et al., 2018; Chu et al., 2019; Bousiotis et al., 2021), and this phenomenon appears to be connected with both regional cloud condensation nuclei production (e.g. Peng et al., 2014; Petäjä et al., 2022) and urban haze formation (Guo et al., 2014; Kulmala et al., 2021, 2022b). The various impacts of atmospheric NPF events depend essentially on their frequency of occurrence and intensity, the latter determined by time-averaged particle formation and growth rates. Quantifying the main

characteristics of NPF in different environments, and connecting these characteristics to emissions and atmospheric processes, is therefore vital for our better understanding on the potential influences of atmospheric NPF on air quality, climate and weather.

Field measurements have been, and will likely continue to be, the primary source of data on the occurrence and intensity of atmospheric NPF (Kerminen et al., 2018; and references therein). Traditionally, the most common way to estimate the NPF event frequency from atmospheric observations is to classify individual measurement days into a small number of categories, from which one then calculates the fraction of days during which NPF events occurs (Dal Maso et al., 2005; Kulmala et al., 2012; Dada et al., 2018). Such NPF event classification methods, while widely

applied in the scientific literature, tend to be subjective and time consuming, and often result in a large fraction of days for which it is difficult to estimate whether a NPF event took place or not. The subjectivity and time consumptions issues can be alleviated using automatic or semi-automatic methods applied to field measurement data (e.g. Joutsensaari et al., 2018; Zaidan et al., 2018; Su et al., 2022), but the problem of having a larger fraction of days difficult to classify tends to remain.

An alternative way to approach the NPF event frequency is to define some indicator, based on quantities obtained from field measurements, that predicts NPF events in a more probabilistic way (Hyvönen et al., 2005; McMurry et al., 2005; Kuang et al., 2010; Jayaratne et al., 2015; Cai et al., 2021; Olin et al., 2022). The benefits of such approaches compared to the traditional NPF event classification methods are that they are usually faster and more easily applicable to all measurement days. However, the indicators developed so far tend to be sensitive to the dominant NPF pathway

and possibly other site-specific factors, requiring a priori knowledge of the mechanisms involved in NPF at any given site or requiring complementary measurements, such as concentration measurements of sulfuric acid and other precursors gases that are not always available in the field.

In the vast majority of studies, the intensity of NPF was estimated only for days showing clear signs of both particle formation and subsequent particle growth. This is an undesirable feature, since atmospheric NPF appears to proceed

on other types of days as well, albeit typically with weaker intensities (Kulmala et al., 2022a). Currently available tools have a hard time in quantifying these weak-intensity, yet often non-negligible, periods of NPF because of instrumental limitations and heterogeneities in measured air masses, limiting determination of both particle formation and growth rates.

In this manuscript, we present a novel approach, the Nano Ranking Analysis, for characterizing NPF from atmospheric

observations. In the following sections, we begin by introducing and detailing the Nano Ranking Analysis. Subsequently, we offer an overview of the measurement site and the instruments employed in our study. Finally, we outline the procedure for calculating the formation rate, as well as the utilization of the traditional classification method. Both components will then be used to demonstrate the effectiveness of our novel approach.

## 2 Methods

### 2.1 Description of the Nano Ranking Analysis

The Nano Ranking Analysis is designed to characterize NPF events in an objective, quantifiable and replicable manner. Our foundational supposition, which is in alignment with earlier observations of atmospheric ions (e.g. Tammet et al., 2014; Leino et al., 2016), asserts that the daily fluctuation of the particle number concentrations within the 2.5–5 nm diameter range ($\Delta N_{2.5-5}$) is acutely sensitive to the presence of atmospheric NPF. The range is defined with a lower limit of 2.5 nm to prevent interference from the continuous mode of molecular clusters (Kulmala et al., 2007). Moreover, detecting particles at this size inherently indicates that nucleation and growth are occurring in the atmosphere. The upper limit is set at 5 nm to maintain a strong signal while minimizing the impact from primary ultrafine particle sources, such as traffic emissions. These emissions are observed to decrease rapidly in concentration, favouring smaller particle sizes (Rönkkö et al., 2019; Karl et al., 2016, Ketzel et al., 2004). While the specific range can be adjusted based on site conditions and available instrumentation, it is recommended not to exceed 7 nm to avoid potential non-NPF related disturbances in the signal.

Building upon this premise, our method quantifies NPF events using their corresponding $\Delta N_{2.5-5}$ value (formal description in section 2.1.1). This value practically represents the daily difference between the maximum and minimum concentrations of these particles, and conveniently serves as a unique and continuous metric. The approach yields a single representative value for each measurement day. The maximum and minimum values can be constrained to specific time windows denominated 'active' and 'background' periods (as described below), in which these values are anticipated. This helps mitigate potential interference from well-known primary emission sources such as traffic. Subsequently, we employ a two-fold approach: firstly, the derived $\Delta N_{2.5-5}$ values are used to rank NPF events, and secondly, we scrutinize the logarithmic distribution of these values to discern any dominant modes. These modes can be further fitted using Gaussian curves, thereby serving as a useful tool to differentiate between varying intensity levels, or "modes", of NPF. We anticipate that these dual outputs will facilitate a deeper understanding of the mechanisms underpinning NPF for specific sites. The former can be compared with continuous parameters linked to NPF, such as precursor gases, condensation sink, meteorological conditions, and the time over land of the associated air mass. In contrast, the latter can be aligned with categorical elements like synoptic patterns, transport mechanisms, or volcanic eruptions. The modes also make the comparison between different sites more robust as it is possible to, for example, justify the comparison between "intense events" in different environments (i.e., compare their frequency and intensity) for the highest mode. Such a comparison would be more challenging using just a specific numerical threshold ($\Delta N_{2.5-5}$> "certain limit"), as this approach might not account for potential differences in condensation/coagulation sink and other parameters affecting NPF, in addition to NPF itself.

While there is no strict minimum on the number of days required to implement this method, we advise a baseline of one month to increase the likelihood of obtaining a representative sample of NPF intensity and occurrence. However, more extensive datasets are preferable, with multi-year and multi-season time series being optimal.

### 2.1.1 Steps to calculate $\Delta N_{2.5-5}$

The following steps outline our approach for analysing the days based on the $\Delta N_{2.5-5}$ spectrum:

1. *Extract the timeseries of the particle number concentration ($N_{d_1-d_2}$)* in the size interval immediately above the continuous mode of molecular clusters from the particle number size distribution (Fig. 1A-B). In our case we use the particles in the size range of 2.5–5 nm.

2. *Smooth the $N_{2.5-5}$ timeseries* to mitigate the influence of potential spurious signals on the ranking value. In our case, we applied a rolling median over 2-hour intervals. This approach reduces the impact of noise or outliers, ensuring a more reliable and accurate ranking assessment.

3. *Identify diurnal background and active regions*. Background regions are generally characterized by times of day that have minimal diurnal values, whereas active regions exhibit maximal diurnal values of $N_{2.5-5}$. To identify these regions more accurately, we recommend dividing the dataset into seasons and examine the diurnal behaviour in each season separately (Fig. 1C). This is particularly important in environments with high levels of particle emissions (particularly nanoparticle emissions), such as urban environments, in which poorly chosen time regions could result in values of $\Delta N_{2.5-5}$ considerably affected by these emissions.

4. *Find the background number concentration for each day ($N_{B;2.5-5}$)*. The background concentration corresponding to a given day is determined based on the median value of $N_{2.5-5}$ in the so-called background region after applying the 2-hour rolling smoothing of the timeseries (step 2). For Hyytiälä, this time window is between 21:00 and 06:00 (Fig. 1D). The window boundaries are selected so that they contain the seasonal daily median minimums (Fig. 1C). These boundaries are site specific and slightly different values are expected for other sites.

5. *Find the active peak daytime number concentration ($N_{A;2.5-5}$) for each day*. The active peak concentration corresponding to given day is determined based on the max value of $N_{2.5-5}$ in the so-called active region. Note that the $N_{2.5-5}$ timeseries has been previously smoothed to a 2-hour rolling median (step 2), and this will impact the maximum value. For Hyytiälä, the active time window is between 06:00 and 18:00 (Fig. 1D)

6. *Determine the change in number concentration ($\Delta N_{2.5-5}$) for each day*. This value is defined as

$$\Delta N_{2.5-5} = N_{A;2.5-5} - N_{B;2.5-5} \qquad (1)$$

and it is the metric used to characterize the strength of potential NPF occurrence for the corresponding day (Fig. 1D).

7. *Rank the days in percentiles* based on their corresponding value of $\Delta N_{2.5-5}$ and group the days based on 5% intervals (Fig. 2) to assess the corresponding potential NPF pattern of each interval.

**2.1.2 NPF "mode" fitting**

The $\log(\Delta N_{2.5-5})$ distribution can be used to identify NPF modes based on their intensity. The procedure is as follows: first, the $\log(\Delta N_{2.5-5})$ distribution is depicted (Fig. 3A), and a visual assessment is made to determine the number of Gaussian curves needed to describe the distribution— in our case, three curves. Next, the distribution is fitted using three Gaussian functions (g1, g2, g3). Initial guesses for the Gaussian's center, width and amplitude and their allowed range in the fitting algorithm are provided based on visual inspection. The division between these groups is determined by finding the midpoint between the centers of two subsequent Gaussians (Fig. 3, dashed line). The intensity of NPF events is assessed within each group by plotting the diurnal median particle number size distribution (Fig. 4, third

row), so that both visual (Fig. 4, first row) and statistical (Fig. 4, second row) inspections of the diurnal variation of $N_{2.5-5}$ can be performed for each group.

## 2.2 Description of the dataset

### 2.2.1 Site description

All the measurements were conducted at the Station for Measuring Ecosystem–Atmosphere Relations (SMEAR) II, in Hyytiälä, southern Finland (61°51´N, 24°17´E; 181 m A.S.L.; Hari and Kulmala, 2005; Hari et al., 2013). The SMEAR II station is located in a boreal pine forest, in a rural environment. The nearest large city is Tampere, located approximately 60 km southwest of the station, with a population of around 200 000 residents. Additionally, comprehensive observations of trace gases, soil-atmosphere fluxes, as well as meteorological variables, have been

concurrently conducted at the site. More details about the station can be found from Hari and Kulmala (2005).

### 2.2.2 Neutral cluster and Air Ion Spectrometer (NAIS)

We used data from the NAIS (Neutral cluster and Air Ion Spectrometer, Airel Ltd.; Mirme and Mirme, 2013). The NAIS measures the number size distributions of ions and total particles in the electrical mobility diameter ranges 0.8-42 nm and 2.5-42 nm respectively.

The NAIS has two measurement columns operating in parallel, one for each polarity. During ion measurements, the positive and negative ions are simultaneously measured in the two columns. During particle measurements, aerosol particles are charged to opposite polarities using corona chargers and simultaneously measured in both columns. The particle data below about 2 nm is contaminated by charger ions and is not included in the measured size range for particles. Air ions and charged particles are separated based on their electrical mobilities and detected in a

multichannel differential mobility analyzer (DMA).

Particle number size distribution (PNSD) data from the negative polarity was used and the number concentrations of total particles between 2.5 and 5 nm were determined based on interpolation. The number concentrations of 2.5–5 nm particles were used in the Nano Ranking Analysis presented in this study. A visual screening process was performed to identify and remove faulty or rain or snow contaminated measurements from the dataset (11 days).

### 2.2.3 Differential Mobility Particle Sizer

The Differential Mobility Particle Sizer (DMPS) measures both the neutral and charged PNSD (Aalto et al., 2001). A typical DMPS setup contains a bipolar charger to assess the equilibrium charge distribution, a DMA (Differential Mobility Analyzer) to sample particles at specific particle sizes, and a CPC (Condensation Particle Counter) to measure particle number concentrations.


At the Hyytiälä site, the measurements are taken with a twin-DMPS system consisting of two separate DMPSs operated in parallel. The first DMPS determines the size distribution of small particles, 3–50 nm in diameter, and the other measures larger particles, 15 nm–1000 nm in size. Together they cover the size range of 3–1000 nm. The CPCs used for particle counting are a TSI-3025 and a TSI-3010 (TSI Inc.), respectively. Both DMPSs contain a dryer at the

inlet, with a continuous sheath flow with a relative humidity below 30% to enable measuring the particle size under
relatively dry conditions. The measurement height of the DMPS system is at ground level at the station (~2 m), and
the time resolution of the measurements is 10 minutes. The DMPS outputs were inverted following the normal pseudo
inversion routine with DMA kernels (Stolzenburg, 1988). The inverted outputs were used for visualization of the Nano
Ranking Analysis percentile bin separation (Fig. 2), as well as for the traditional NPF event classification and
calculations of particle formation rates.

### 2.2.4 Traditional event classification

We compare the ranks from the Nano Ranking Analysis with the traditional NPF event classification (see section 3,
Fig 7.) introduced by Dal Maso et al. (2005). The traditional event classification categorizes days into NPF event days,
undefined days, and non-event days by visually analyzing the particle number size distribution data from a DMPS on
a day-to-day basis. This classification procedure characterizes NPF events by the growth of a new mode of particles
in sub-25 nm over a time span of hours. Additionally, the classified NPF events are divided into three sub-classes
(event Ia, Ib, and II) based on the level of confidence determined by particle growth and formation rates. A detailed
description of the traditional NPF event classification can be found in Dal Maso et al. (2005).

### 2.2.5 Particle formation rate ($J$) calculations

The particle formation rates ($J_{D_{p_i}}$) describes the flux of growing particles past some diameter $D_{p_i}$ and were calculated
following the scheme described in Kulmala et al. (2012). The formula is shown below:

$$J_{D_{p_i}} = \frac{dN_{[D_{p_i}, D_{p_u}]}}{dt} + \sum_{l=i}^{u-1} \sum_{j=l}^{max-1} K\left(D_{p_l}, D_{p_j}\right) \cdot N_{[D_{p_j}, D_{p_{j+1}}]} \cdot N_{[D_{p_l}, D_{p_{l+1}}]} + \frac{GR_{[D_{p_i}, D_{p_u}]}}{D_{p_u} - D_{p_i}} \cdot N_{[D_{p_i}, D_{p_u}]}$$

(2)

where the first term stands for observed time derivative of particle number concentration calculated from particle
number size distributions (PNSD) from DMPS measurements. In our case we chose $i$ and $u$ such that $D_{p_i} = 3nm$ and
$D_{p_u} = 7nm$. The second term describes the coagulational scavenging which considers the sum of coagulation sinks
of particles from each size bin where $K$ is the coagulation coefficient of particles in the size $D_{p_l}$ and $D_{p_j}$. The sum is
over all bins from the DMPS between $D_{p_i}$ and $D_{p_{max}} = 1000$ nm. A correction factor for particle hygroscopic growth
is applied when calculating coagulation sinks as described by Laakso et al. (2004) who developed a hygroscopic
growth factor for specifically for Hyytiälä as a function of measured relative humidity. This method can increase the
accuracy when estimating coagulation sink of particles over different sizes, where their hygroscopic properties also
differ.

The last term in Eq. (2) accounts for particle losses due to growth into larger sizes. We used the maximum
concentration method described in Kulmala et al. (2012) to calculate the particle growth rate $GR_{[3,7)}$ from days that
were classified as having NPF events (event Ia, Ib and II) based on the guideline illustrated in Dal Maso et al. (2005)
which is briefly discussed in the previous section. Since determining individual growth rates for non-event days is not

possible with the existing methods, we determined the GR for non-event days by calculating a median $GR_{[3,7]}$ based on the method described by Kulmala et al. (2022b). This method first takes the diurnal median PNSD for all non-events and then normalizes each size bin by the maximum value of that bin, thus revealing the hidden growing mode for these non-event days (same normalization is used in Fig. 9). This procedure allows the quantification of the growth rate and subsequently the calculation of $J_3$ for each individual non-event day. In short, for non-event days, we follow the same procedure used to calculate event-day $J_3$ except that instead of using the $GR_{[3,7]}$ for a specific day—which cannot be calculated for single non event days—we use the median $GR_{[3,7]}$ of all non-event days. As comparison, the median $GR_{[3,7]}$ for event and non-event days was 5.1 nm h$^{-1}$ and 3.6 nm h$^{-1}$ respectively.

**3 Results of applying the method in the boreal forest**

In this section, we present the findings of our research derived from the Nano Ranking Analysis. We focus on the comparison between the $\Delta N_{2.5-5}$ metric—either directly or via its ranks—and traditional parameters in the study of NPF. Specifically, we examine the comparison of $\Delta N_{2.5-5}$ and diurnal $N_{2.5-5}$ concentration patterns (Fig. 5), new particle formation rates (Fig. 6), traditional classification of events (Fig. 7), and seasonality (Fig. 8). Finally, we apply the normalization proposed by Kulmala et al. (2022) to reveal quiet NPF patterns in the percentile interval bins (Fig. 9). These results provide valuable insights into the effectiveness and applicability of our proposed metric in capturing key aspects of NPF dynamics.

Figure 5 shows the median particle number concentrations $N_{2.5-5}$ for different hours of the day and for different intensity rank values. We can see that the concentration profile throughout the day is relatively similar for rank values approximately below 50 % with little variation in $N_{2.5-5}$. However, for ranks above 50 % slight increases in $N_{2.5-5}$ during daytime hours are observed. As the rank increases, values of $N_{2.5-5}$ are correspondingly higher. In addition, increased daytime particle concentrations last longer for days with higher intensity ranks. For example, for rank values of 85–90 % the increased concentrations are mainly present between 12:00 and 18:00, while for rank values of 95–100 % they last two to three hours longer, from approximately 11:00 until 19:00–20:00. Fig. 5 shows how the analysis method presented can be used to study the particle concentrations as well as hours of the day during which NPF potentially takes place.

Figures 6A and 6B show the particle formation rate, $J_3$, as a function of $\Delta N_{2.5-5}$ and of the percentile ranking, respectively. The value of $J_3$ clearly increases with increasing $\Delta N_{2.5-5}$ and, as a result, with an increasing percentile ranking. $J_3$ is the highest in group g3 and the lowest in group g1. In group g1, corresponding to percentile rankings smaller than about 70 %, average values of $J_3$ slowly increase with increased rank. In group g2, and especially in group g3, the increase of $J_3$ with increasing rank is stronger. The majority of cases with $J_3$ larger than 0.1 cm$^{-3}$s$^{-1}$ occur when the percentile ranking is larger than 80 %, with these values mostly belonging to groups g2 and g3. Therefore, Fig. 6 illustrates the clear connection of ranking with intensity of atmospheric NPF. If the percentile rank of the day is high, the intensity of NPF can also be expected to be relatively high.

Previously, studies of continuous variables and their effect on NPF have been constrained by a binary division of days into NPF event and non-event days. Figure 6 demonstrates that the novel Nano Ranking Analysis presented here can

be compared with continuous variables, such as the particle formation rate, in investigating NPF. In this case we use the day's maximum $J_3$ and compare it to its corresponding $\Delta N_{2.5\text{-}5}$ (fig. 6A) and percentile ranking (fig. 6B). For the location and data set considered here, this daily $J_3$ maximum in general corresponds to a time window similar or very close in time to the time when the max $N_{2.5\text{-}5}$ (fig. 1D) is found for each individual day. It should be noted that in addition to NPF, the value of $\Delta N_{2.5-5}$, and thus, the intensity rank, can be affected by factors such as coagulation sink, other sources of sub-5 nm particles, and ion or particle production associated with e.g., rain. In our case, the days included in our analysis were visually screened for precipitation to account for this. As a result, while the presented method characterizes NPF intensity statistically, some individual days might have lower NPF intensity than their corresponding rank would suggest.

Figure 7 shows the distribution of different NPF event classes, as per the traditional classification scheme, for the different percentile rankings. The number of days classified as non-events is the highest when the rank is close to zero, and most of the days with ranks below 50 % are classified as non-event days. The number of non-event days decreases with an increasing rank value, such that only a couple of days above the percentile rank of 90 % are classified as non-event days, representing a marginal fraction of all the days. The fraction of days classified as NPF event days starts to clearly grow after the percentile ranking is over 65 %. However, in Hyytiälä most days are classified as NPF event days only for percentile rankings larger than 85 %. In conclusion, Fig. 6 illustrates the connection of the proposed ranking method with the intensity of particle formation and Fig. 7 illustrates its connection with the probability of the occurrence of atmospheric NPF events.

Figure 8 highlights the benefits of utilizing a continuous variable, such as $\Delta N_{2.5-5}$, to characterize new particle formation. This continuous nature enables the application of techniques like a rolling median filter, which is exclusive to non-categorical variables. In the figure, each red point represents a day, while the blue curve results from a 30-day rolling median. This representation distinctly reveals the seasonality of NPF occurrence, indicating a heightened intensity and/or occurrence probability during spring. Employing a categorical classification for NPF events would complicate this analysis considerably. Lastly, as illustrated in Figure 9, the combination of the normalization method proposed by Kulmala et al. (2022a) and the percentile bin intervals (as shown in Fig. 2), significantly enhances the visibility of NPF event patterns, even in ranges that precede those depicted in Figure 2. A well-defined NPF event pattern becomes observable starting from the rank of about 55 %, with a less obvious pattern discernible starting from the ranks of about 25 %. Moreover, the observation that each successively higher interval reveals an increasingly distinct NPF event pattern strongly suggests that the Nano Ranking Analysis effectively orders these events, even when they initially seem indistinct.

Finally, the modes derived from the distribution of $\log(\Delta N_{2.5-5})$ may indicate distinct sources, including regional, meteorological, or emissions-related factors. It is crucial to contrast these findings with available precursor gases, transport patterns, and meteorological variables. Further studies applying this methodology in various scenarios will provide insights into, and guidance for, effectively implementing this new metric.

**4 Considerations and limitations of the method and its applicability to other sites**

Now we address potential limitations of the proposed NPF metric ($\Delta N_{2.5-5}$). It is important to acknowledge that the high values in this metric could be influenced by factors such as pollution, or by ions or particles produced by either rain (Wimmer et al., 2018) or blowing snow (Chen et al., 2017). On the other hand, a similar formation rate of new

particles will result in a lower concentration of 2.5-5 nm particles when the coagulation sink is higher, as a larger fraction of the formed particles are scavenged by the pre-existing particles (e.g. Kerminen and Kulmala, 2002; Lehtinen et al., 2007). Consequently, an event with a high NPF ranking may not necessarily correspond to an atmospheric NPF event (in case of rain or other particle sources) and, likewise, both intensity and rank of any NPF event may be influenced by varying background aerosol conditions. However, no NPF event classification scheme or

metric method is entirely exempt from errors, and the observation of NPF events is always dependent on measurement instruments (their size range and noise level) and whether the formation rate is high enough to produce an observable increase in particle concentrations over the influence of coagulation sink and other sub-5 nm particle sources. Depending on the dataset, one may want to filter out such influencing factors before applying the Nano Ranking Analysis for example by manually removing all days with rain (as done in this study) or blowing snow or by carefully

choosing the background region so that the effects of polluted primary sources in the 2.5-5nm range are diminished. The application of our new method to different scenarios requires understanding of the specific dynamics unique to each site. Variations in the time windows of active and background regions are expected, and these might even vary within the same site depending, e.g., on the time of the year. Thus step 3 (fig 1C) is very important. Additionally, the metric utilized in this study (Eq. 1) may not be optimally suited for other sites. For example, in environments with

high pollution, the background concentration of $N_{2.5-5}$ may consistently remain elevated. In such cases, it may be more appropriate to calculate the ratio between the active and background time windows rather than simply subtracting them. The method will likely not work at roadside or curbside sites as the sub-5nm population is highly sensitive to traffic emissions at these distances (Rönkkö et al., 2017). However traffic influence on the smallest particle sizes diminishes exponentially with distance from these sources and at distances larger than 100 of meters the number

concentration of these particles is very similar to the background (Rönkkö and Timonen, 2019 and references therein). Furthermore, in urban background environments, ultrafine particle pollution peaks stemming from traffic emissions and those linked to new particle formation (NPF) tend to occur at slightly different times—traffic emissions peak in the morning, while NPF peaks around midday (Trechera et al., 2023). Therefore, a thoughtful selection of both active and background regions, coupled with a reasonable distance of several hundred meters from traffic sources, is likely

to enhance the effectiveness of applying this method in these environments.

Finally, the specific range can be adjusted based on site conditions and available instrumentation, it is recommended not to exceed 7 nm to avoid potential non-NPF related disturbances in the signal. A sensitivity analysis of different size range combinations and reduction functions against classical event classification is provided in fig. S1.

**5 Conclusions**

Formation of fresh atmospheric aerosol particles is a worldwide phenomenon. Here we present a new method to analyze NPF events. Instead of traditional binary NPF event day vs non-event day analysis, we have developed the Nano Ranking Analysis, which ranks the days based on the concentration of 2.5–5 nm particles seamlessly from very low values to high ones. At the same time, the frequency, or mode, of high-ranking values – earlier referred as clear NPF event days – and low-ranking values (clear non-event days) can be obtained.

The new Nano Ranking Analysis is an automatic and objective way to characterize two important aspects of NPF events, namely their intensity and occurrence probability. This method integrates the traditional analysis of investigating NPF on NPF event days and the approach by Kulmala et al. (2022b) investigating NPF on the days when a NPF event is not observed. The days with high (low) ranking values show typically higher (lower) particle formation rates.

Our finding enables new ways to investigate connections between different days. This includes studying how factors such as vapor concentrations, precursor gases, condensation sink, meteorology and transport mechanisms impact the intensity and occurrence probability of NPF. In the future it will be important to investigate connections of the $\Delta N_{2.5-5}$ metric—or its ranks—with atmospheric conditions in different environments. While testing this method in other environments is not yet completed and will be presented in other publications, we envision that the new method is

applicable to many other types of environments and, besides providing probabilistic information about the occurrence and intensity of NPF, it has potential to provide valuable insight into the origin of newly formed particles at sites affected by multiple sources of aerosol precursors.

**Code availability**

- Code for the data analysis performed in this study can be downloaded from the following repository hosted
at GitHub (https://github.com/daliagachc/ranking-hy_zenodo) and an archive of the repository can also be found at https://zenodo.org/doi/10.5281/zenodo.10231522 (Aliaga, 2023a)

- The dataset used for this study can be downloaded from https://zenodo.org/doi/10.5281/zenodo.10231189 (Aliaga, 2023b)

**Author contribution**

MK conceptualized the original idea. DA developed the method with help from MK, VMK and PP. DA performed the analysis with help from ST, TZ, JL, XL, and SH. JL and TN collected the data. DA prepared and edited the manuscript with contributions from all co-authors.

**Competing interests**

MK is member of the editorial board of the Aerosol Research journal. The authors have no other competing interests to declare.

**Acknowledgements**

We acknowledge the following projects: ACCC Flagship funded by the Academy of Finland grant number 337549 (UH) and 337552 (FMI)**;** Academy professorship funded by the Academy of Finland (grant no. 302958); Academy

of Finland projects no. 1325656, 311932, 334792, 316114, 325647, 325681, 347782, 337549; "Quantifying carbon sink, CarbonSink+ and their interaction with air quality"; INAR project funded by Jane and Aatos Erkko Foundation; "Gigacity" project funded by Wihuri foundation; European Research Council (ERC) project ATM-GTP Contract No. 742206; European Commission, H2020 Research Infrastructures (CHAPAs grant no. 850614; FORCeS grant no. 821205), and Horizon Europe Infrastructures (FOCI grant no. 101056783). Doctoral School in Atmospheric Sciences

at the University of Helsinki (ATM-DP) is acknowledged. University of Helsinki support via ACTRIS-HY is acknowledged. Support of the technical and scientific staff in Hyytiälä are acknowledged. ChatGPT (GPT-3.5;4, OpenAI's large-scale language-generation model) has been used to improve the writing style of some excerpts in this article. DA reviewed, edited, and revised the ChatGPT generated texts to his own liking and takes ultimate responsibility for the content of this publication. DA acknowledges the valuable ideas provided by Myriam Agrò and

Sasu Karttunen regarding the evaluation of the method's performance. DA wants to thank Sara Blichner for her thoughtful comments and support during the writing process.

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

**Figures**

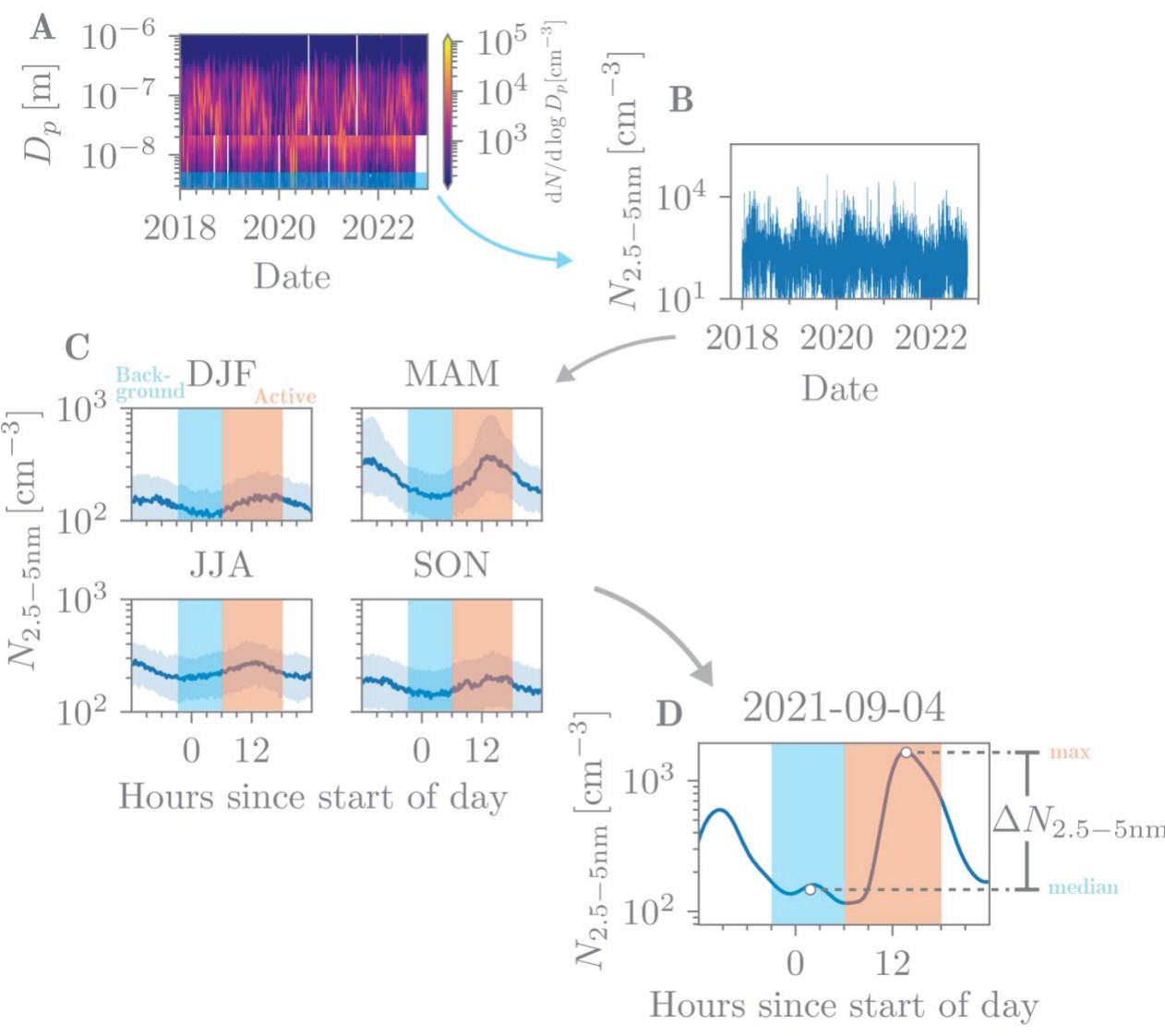

**Figure 1: Diagram illustrating the process used to calculate the $\Delta N_{2.5-5}$ metric. The methodology involves extracting particle concentrations in the 2.5 to 5nm range (B) from the particle number size distribution timeseries (A), followed by grouping the timeseries by season and plotting daily patterns identifying the background and active zones (C). The final step calculates the daily difference for each day—here exemplified by the 4th of September 2021— between the maximum concentration in the active region and the median concentration in the background region (D).**

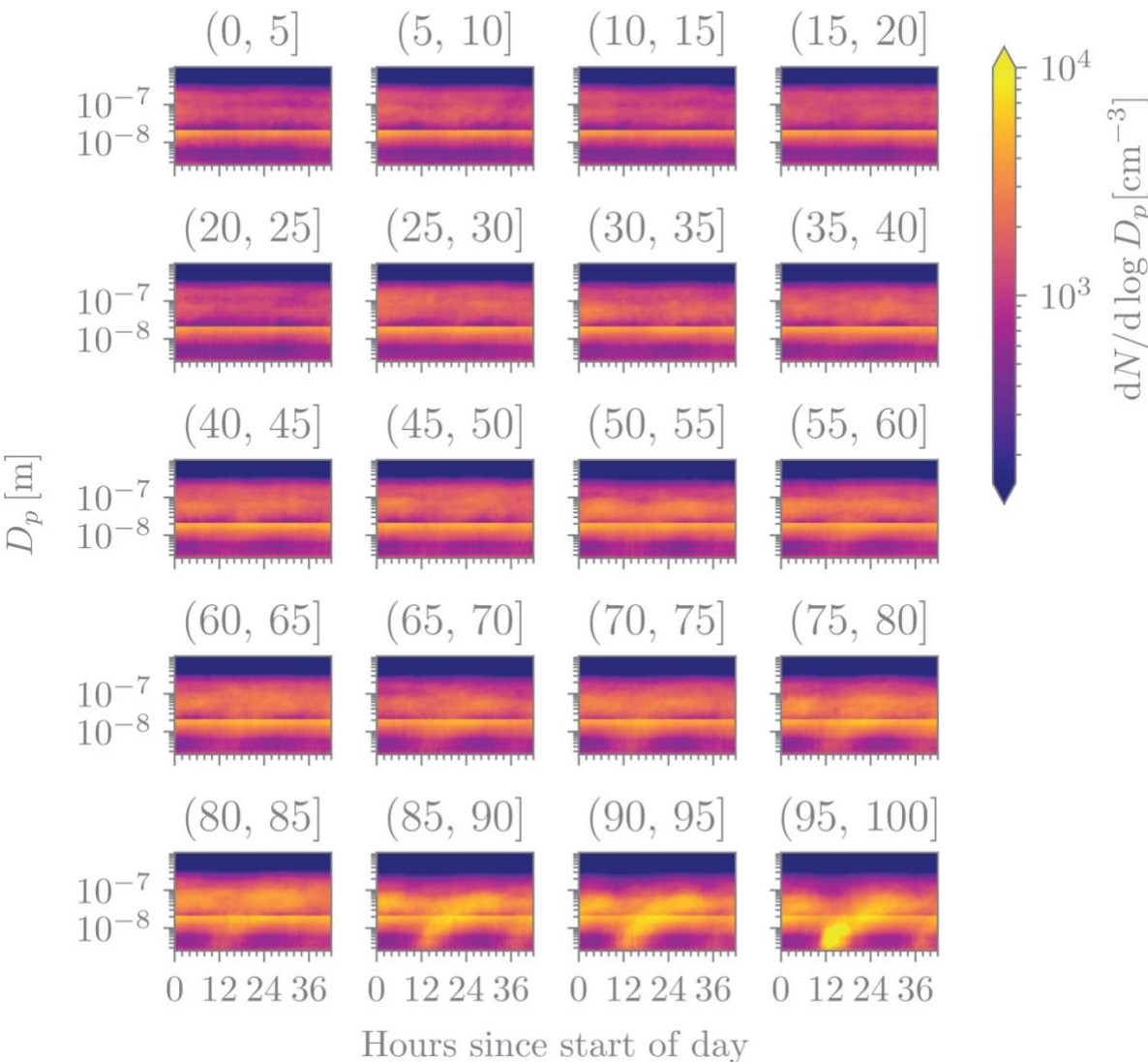


**Figure 2: Daily median number particle size distribution grouped into 5% intervals based on the $\Delta N_{2.5-5}$ ranks. The diameter limits (y-axis) in the surface plots are 2.5 and 1000 nm and are used to illustrate the shape of the potential NPF events in each interval. For illustration purposes, the particle number concentrations obtained from NAIS (2.5-20nm) and DMPS (20-1000nm) are combined.**

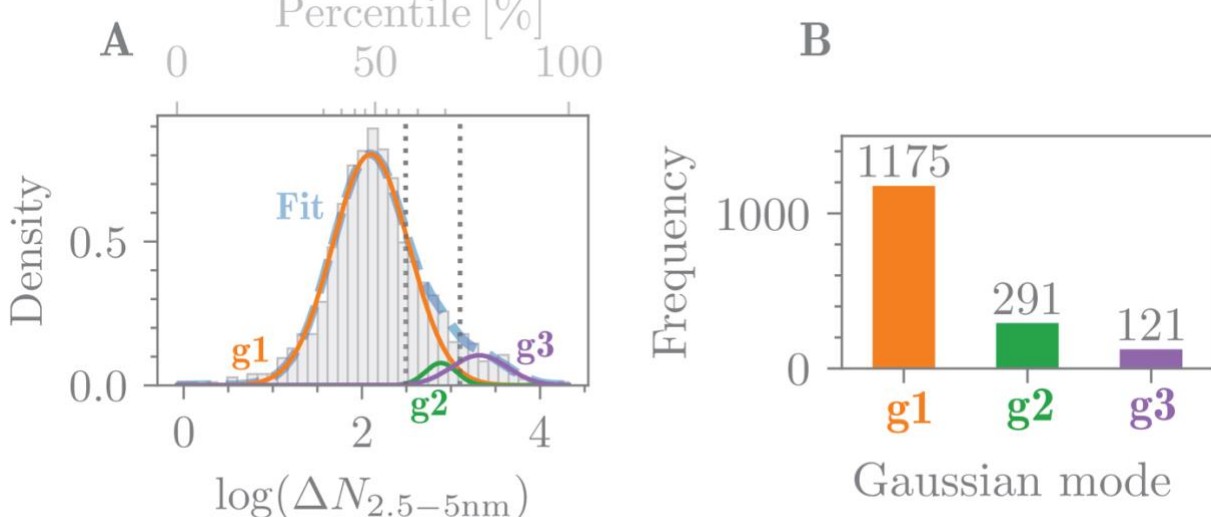

Figure 3: (A) Density histogram (gray) based on the daily log($ΔN_{2.5-5}$) values. Additionally, three Gaussian curves (g1, g2, g3) are fitted (sky blue) to the distribution. (B) Number of days belonging to each of the identified Gaussian curves in (A). The regions corresponding to each group are determined by calculating the middle distance (dashed lines) between adjacent Gaussian curve centers.

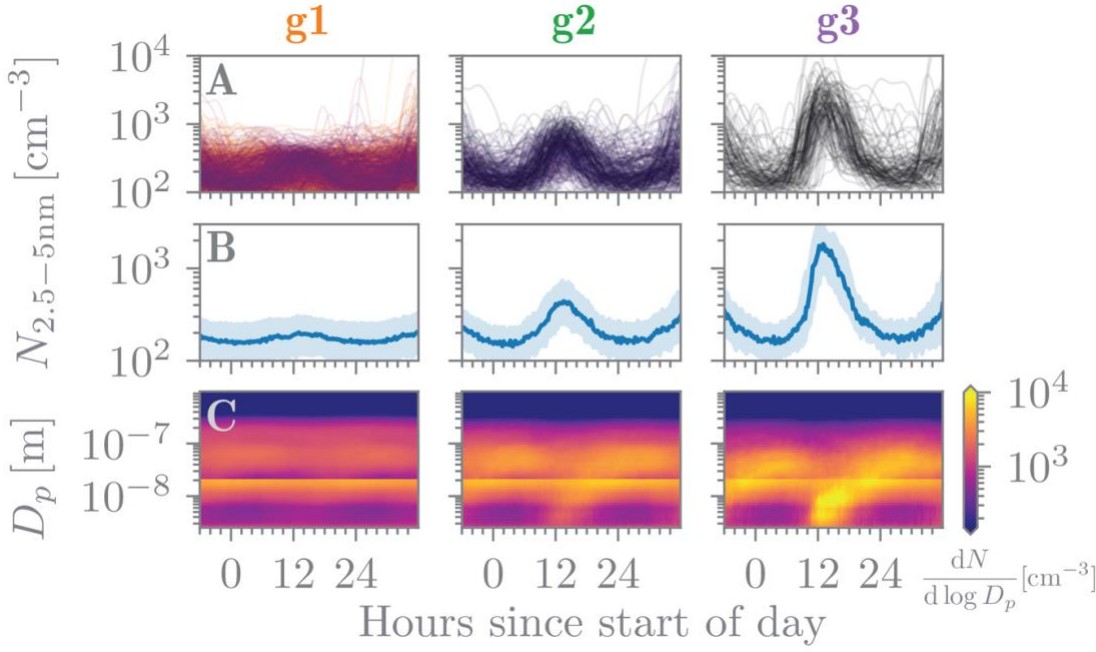

Figure 4: The first row displays the daily curve of particle concentrations in the range of 2.5 to 5nm, grouped into three columns: g1, g2, and g3. The second row shows the median and interquartile range of the daily curves shown in the first row. The third row presents the median particle number distribution for the days belonging to each group g.

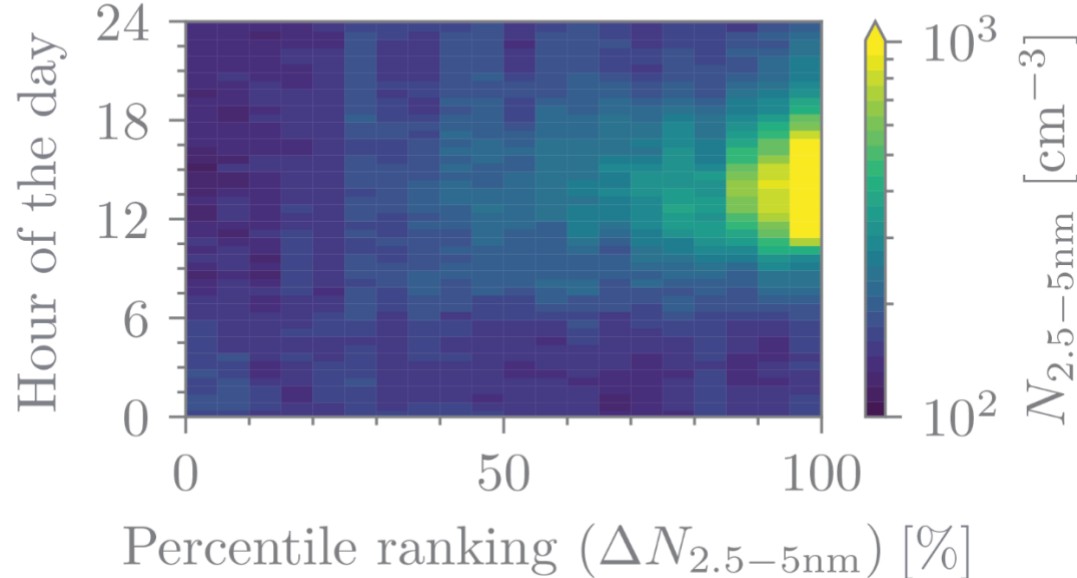

**Figure 5: Median 2.5-5 nm particle number concentrations (N$_{2.5-5\ nm}$) for different times of the day and for different percentile ranking values, which were based on $\Delta$N$_{2.5-5\ nm}$. The percentile rankings have been divided into 5% intervals, while hourly time resolution was used.**


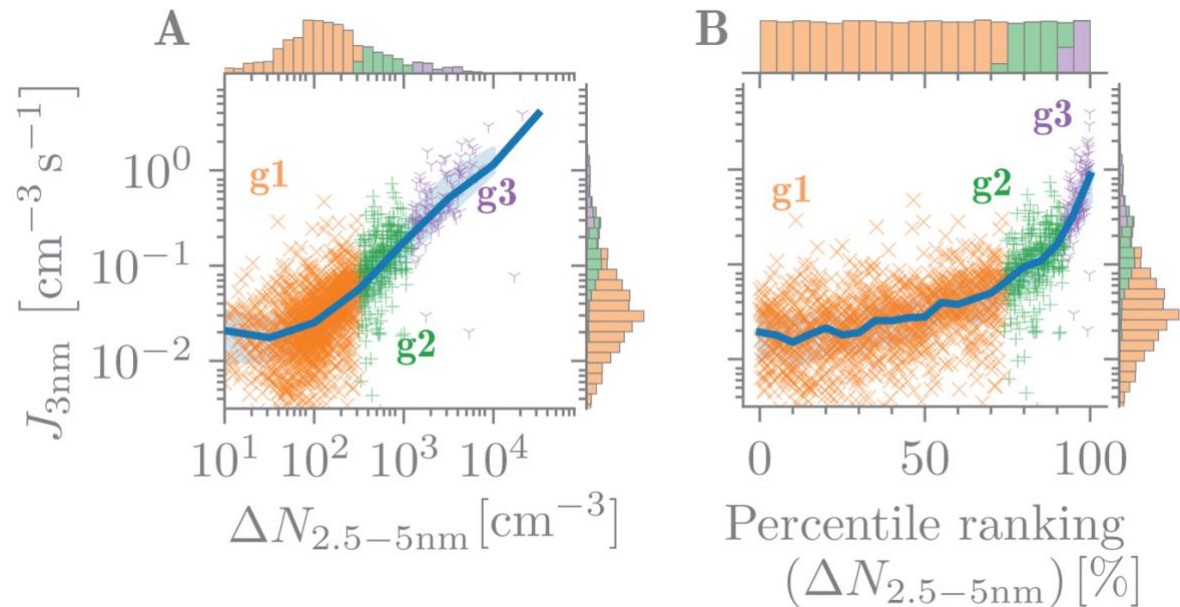

**Figure 6: (A) Daily $\Delta N_{2.5-5}$ (x) vs J (y) color-coded by group g. The blue line represents the median, and the shaded region indicates the interquartile range. At the top of the panel, a histogram of $\Delta N_{2.5-5}$ values is presented, while on the right side, a histogram of J values is shown, both color-coded by group g. (B) Similar to panel A but using percentile ranking instead of $\Delta N_{2.5-5}$.**


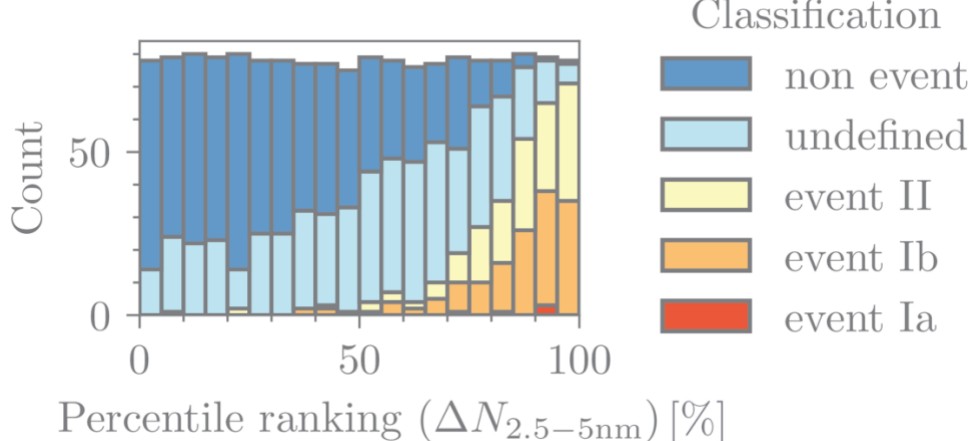

**Figure 7: Comparison between percentile ranking and traditional classification. This histogram displays the percentile rankings divided into 5% bins and color-coded based on the classification: Blue (non-event), light blue (undefined), yellow (event II), orange (event Ib), and red (event Ia).**

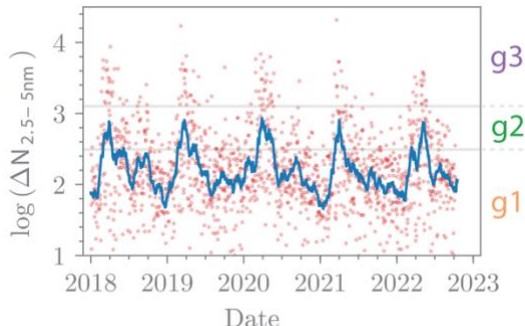

**Figure 8: Daily time series of $\Delta N_{2.5-5}$. The red dots represent the daily values, while the blue line shows the 30-day rolling median. The shaded areas indicate the regions corresponding to the modes g1, g2, and g3.**

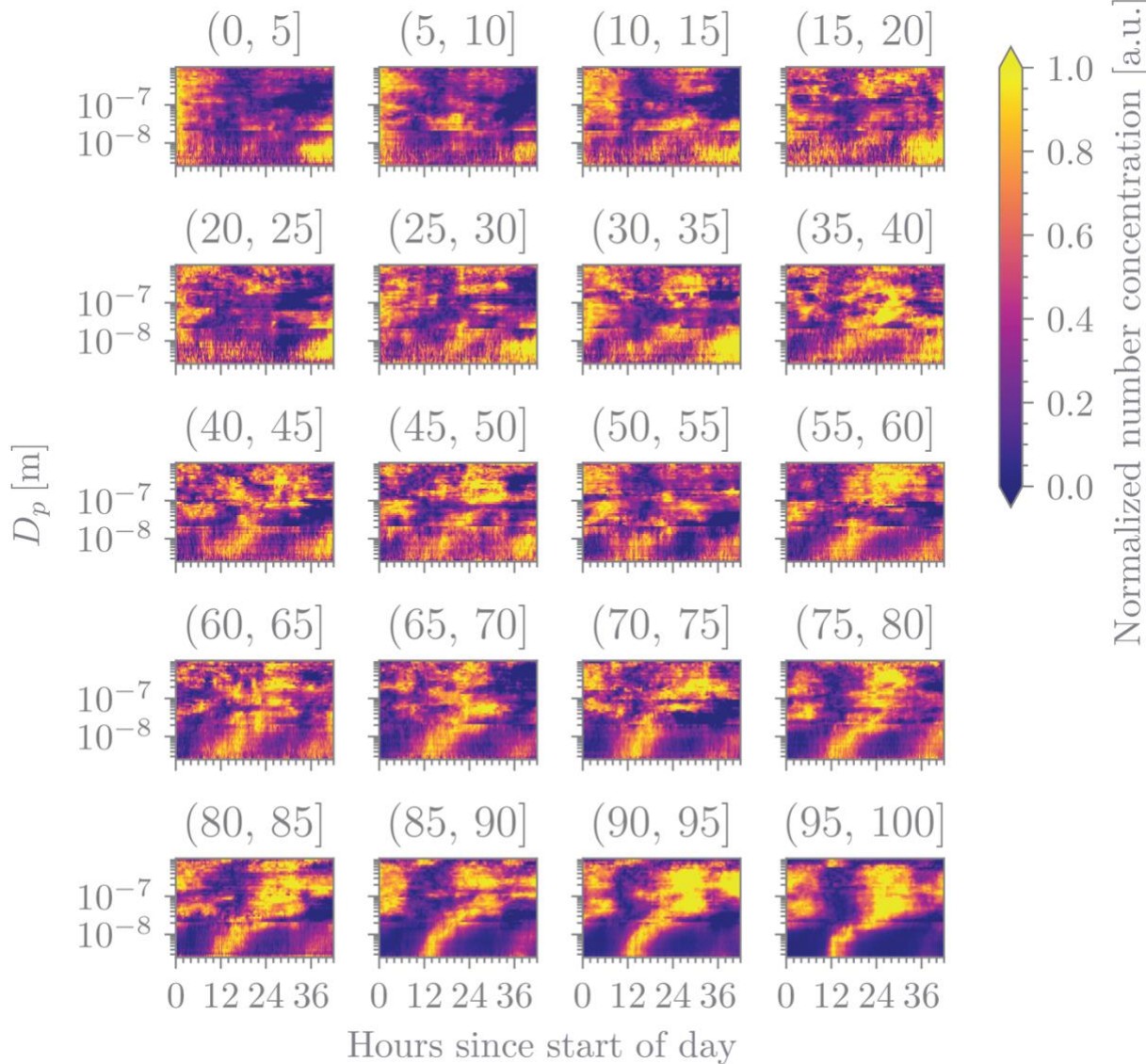

**Figure 9: Similar to Fig. 2 but using the normalization proposed by Kulmala et al. (2022a). In short, for every 5%-interval, each size bin is linearly scaled based on its median maximum and minimum so that values span from 0 to 1.**
