# Peer review of "Nano Ranking Analysis: determining NPF event occurrence and intensity based on the concentration spectrum of formed (sub-5 nm) particles"

_Aerosol Research, 2023_

## Referee Comment (RC3)

The study published by Aliaga et al. concerns the development of a method for automatically providing information on the formation of new particles from observations, both in terms of the occurrence of the process and its intensity. Although its effectiveness is only demonstrated for the boreal forest environment in this study, this method appears to be an interesting alternative/complement to traditional methods that rely on visual detection of events (and are therefore subject to a certain subjectivity); in particular, it overcomes the limitations/difficulties associated with these methods with regard to the identification and characterization of very low-intensity events. I therefore recommend the publication of this work, which I hope will subsequently call for further studies to assess how this approach can be used in environments with contrasting characteristics. The few questions and comments listed below should, however, be considered before publication of this manuscript.

P2, L53-55 : « *However, the indicators developed __so far__ tend to be sensitive to the dominant NPF pathway and possibly other site-specific factors, casting doubts about their general applicability to different atmospheric environments* » : I find this wording somewhat disturbing, as it implies that, unlike existing methods, the proposed method is not site-specific... but this is the case, at least as far as the determination of active and background regions is concerned (as clearly mentioned in Sect. 3). Furthermore, there is nothing in this study to demonstrate the applicability of this method (which limitations are actually discussed at the end of Sect. 3) to other environments (e.g. mountain sites affected by complex boundary layer dynamics; urban areas affected by pollution, with background regions possibly changing on a weekend / week day basis).

P2, L63-64 : About the plan. Since the description of the method is based on the data, I would suggest putting the description of the site and the instrumental set-up before the description of the method. Furthermore, I think Sect. 3 could be split into two sub-sections to improve the clarity of the messages: one associated with the results in the boreal forest, the other dedicated to the limitations of the method / discussion of its use in other environments.

P2, L68 : About the method in general. For me, a few lines clearly explaining how the method can/should be used are missing. I understand that the information associated with ranking and the determination of the modes first requires a sufficiently long data set, and that the "reference image" resulting from this analysis can then be used to "characterize" new data/individual events, to position them. Is this correct? If so, what are the constraints for defining the reference dataset (length, representativeness of different seasons, etc.)?

P5, L146-148: "*The CPCs can detect particle number concentrations ranging from 3 nm to 10 nm, whereas the detection of particle size range for the DMPS is 3–1000 nm.*" The wording does not seem clear to me. I think it should be clear what the cutoff diameter associated with each of the two CPCs is, and then the size range covered by the setup that combines the two CPCs / DMAs.

P5 : About the coagulation sink.

- L167-168 : "*The second term describes the particle losses due to coagulation to larger size particles __with__ corresponds to their number concentrations (NDp).*" Check the wording.
- Equation 3 : I do not understand the proposed formulation if Dp1 and Dp2 have fixed values; in other words, what does the sum relate to?
- L172: "*The correction factor for particle hygroscopic growth is applied when calculating coagulation sinks.*" Can the authors say a little more about this correction factor, or at least cite the study describing its calculation?

P6, L183: "*we examine the comparison of ΔN2.5-5 and diurnal patterns*". The variables concerned by the diurnal variations studied should be indicated.

P6, L204 – 206 : "*In addition, Fig. 6 demonstrates that the novel Nano Ranking Analysis presented here can be compared with continuous variables, such as the particle formation rate, in investigating NPF.*" Could the authors elaborate a little more on how the comparison is carried out? How is the value of J included in the comparison obtained: is it a value (mean, median?) calculated over the whole day, over an identified time window?

Minor / technical comments

P1, L 34 : *Kulmala et al., 2022b* : This paper should be referenced as Kulmala et al. 2022**a** since there are no other references by Kulmala et al. published in 2022 mentioned before.

---

## Author Comment (AC1)

**Table of Contents**

Referees' comments are visually distinguishable by being coloured in blue.

Author's responses are presented in black text.

**1 RC 1**

The study introduces a new and easily applicable metric that can be used to determine the occurrence and estimate the strength of atmospheric NPF. This metric can be very useful as the extraction of NPF events is a laborious and time-consuming task with great uncertainties which come from either from the variable intensity of the events as well as personal perspective.

We thank RC1 for their constructive comments and appreciate that the reach of our study is understood.

**1.1 Major comments**

While the metric can be useful in some cases there are some significant limitations with its design and application.

**1.1.1 The most important one is the complete exclusion of the growth of the particles in considering an event. According to the widely accepted definition of NPF events (Dal Maso et al., 2005) the new mode of particles formed should present signs of growth. This though cannot be determined with the metric presented as the size range chosen is too small for the growth of the newly formed particles to be confirmed. Additionally, without knowing whether there is growth of the particles, there is no information whether the NPF events extracted are atmospherically relevant (affect the local air quality, new particles to become CCN etc.) or if the particles are quickly lost by mechanisms such as coagulation or condensation. This very important aspect is not discussed at all within the manuscript.**

We acknowledge that our proposed metric does not take into account growth in the traditional sense set by Dal Maso et al. (2005). When that classification was developed, the capability to detect particles below 10 nm was limited. More recent methods, like Dada et al. (2018), use concentration of particles or ions as small as <4 nm.

However, our deviation from the traditional approach towards 'growth' is intentional. Observing NPF-driven particle growth in the Aitken mode in the field is complex due to confounding factors like transport dynamics and primary emissions. Under ideal conditions, where particle formation is regionally uniform without interference from primary particles, Dal Maso et al. (2005)'s metric would be optimal. However, real-world atmospheric conditions are more variable, and our metric is designed to address these complexities.

Furthermore, our metric does take into account particle growth in the following way: by starting our range at 2.5 nm, as opposed to the initial clustering size of roughly 1.5 nm, we are inherently capturing the combined effect of nucleation and growth as both need to have happened in order to observe these particles. This is also explicitly mentioned by Dal Maso et al. (2005), albeit for a larger size. The upper limit of our metric is set at 5 nm, but this is not fixed, and slightly larger sizes can be used to increase confidence in the signal (fig. E can be used as a guideline). However, as we consider larger particles sizes, their lifetime in the atmosphere also increases,

introducing potential errors, especially near sources like traffic or industry (Ketzel and Berkowicz, 2004).

In section 2.1 of the text, we have included the following sentences clarifying our approach on growth:

"Moreover, detecting particles at this size inherently indicates that nucleation and growth are occurring in the atmosphere. The upper limit is set at 5 nm to maintain a strong signal while minimizing the impact from primary ultrafine particle sources, such as traffic emissions. These emissions are observed to decrease rapidly in concentration, favouring smaller particle sizes (Rönkkö et al., 2019; Karl et al., 2016, Ketzel et al., 2004). While the specific range can be adjusted based on site conditions and available instrumentation, it is recommended not to exceed 7 nm to avoid potential non-NPF related disturbances in the signal."

Furthermore, our results in this article confirm that high $\Delta N_{2.5-5}$ values usually lead to further growth, likely because the conditions that result in high particle counts in the 2.5-5 nm range typically lead to further growth. Figure A, supports this, indicating a correlation between concentrations in the 2.5-5 nm range and larger sizes later in the day. Furthermore, when events are categorized, either by percentiles (Fig. 2) or by modes (Fig. 4), the evidence of growth becomes clearer. Especially for events in the upper 75th percentile, significant growth is evident, aligning with the data in Figure 3A.

> 1.1.2 The metric (according to the Abstract) is considered to help in determining the occurrence of a NPF event. Looking at figure 7 though, there is about 50% uncertainty up until the percentile ranking of 70%. With only a fraction of the days surpassing that ranking (and still with significant uncertainty), how can the occurrence of an event be determined with confidence, especially for lower rankings? Once again the possible events will be considered as undefined.

Thank you for highlighting this concern regarding our metric in relation to Figure 7. We realize the need for precision in our abstract's language. Specifically, our approach is designed to determine both the likelihood and intensity of a NPF event. We have adjusted our abstract to reflect this and reads as follows:

"… we can determine the occurrence probability and estimate the strength of atmospheric NPF events …"

Regarding Figure 7, it is shown to contrast the traditional classification with our metric. Where the typical method might label a day as "undefined," our metric measures its potential intensity. This differentiation becomes relevant when we consider events that may have varying atmospheric significance. For example, a traditionally "undefined" day with minimal intensity might have limited atmospheric impact, whereas an "undefined" day with significant intensity could be more consequential and is convenient to statistically include it for further analysis rather than discarding it.

We understand that there is uncertainty about whether an event really happens when we look at the lower rankings and compare them to the classical classification. But our main point is not just about whether an event happens or not. Our results show that

for days with lower rankings, if there is an event, it is usually not as strong as the events on days with higher rankings.

1.1.3  Hyytiala is a rural background site far from most of the sources of particles in the size range 2.5 - 5 nm. It is expected that the metric proposed will perform a lot better in such an environment. But for such an environment even a simple observation of either the total number concentration of particles or the particle number concentration in the range 2.5 - 5 nm (without looking at the background and active regions) would provide a similar outcome (in figure 4 the $N_{2.5-5nm}$ seems rather stable for the non-event days). What is the added value of the specific metric? Please discuss.

Thank you for bringing this to our attention. We acknowledge that we may not have emphasized this point sufficiently in the text.

We recognize that Hyytiälä is a thoroughly studied rural background site, making it an exemplary location to trial our new method. Nevertheless, even in such a rural background location, metrics like the total particle count (2.5-1000nm) yield notably different outcomes compared to Nano Ranking. To investigate this, Fig. C shows a comparison of the Nano Ranking (x-axis) against 6 alternative metrics: Max, Median and Mean for the size 2.5-5 nm and 2.5-100 nm, respectively (see figure caption for full description). The alternative approaches clearly produce different ranking values to those found with the Nano Ranking. Max,2.5-5 (fig. Ca) produces the closest results to the original Nano Ranking. The discrepancy between the metrics is in particular showcased in Fig. Cd, where we compare the rankings derived from the maximum daily total concentration (y-axis) against those from Nano Ranking (x-axis). Similarly, relying solely on the daily max 2.5-5nm also yields contrasting results (Fig. Ca).  In figure D, we also compare these 6 alternative ways of producing the ranking against the classical classification (Dal Maso et al. (2005)) where again Max,2.5-5 performs best amongst the alternatives—fewer events appearing in lower ranking and more events appearing in higher rankings.

In figure E we go one step further and compare the cumulative sum of events (Ia, Ib, II) against 12 different ways of obtaining the ranking with varying particle boundaries and functions to obtain the single day value (see figure and figure caption for details). The assumption here is that Dal Maso classification is the ground truth. The effectiveness of a set of parameters is measured by the distribution of events across the ranking spectrum, with the ideal method showing fewer events at lower rankings and a concentration of events at higher rankings. The options in the first four rows are strong candidates. The fifth row's method, which considers particles up to 30nm in diameter, is less ideal, as it begins to register events at very low-ranking values (around 15%). The sixth option, which takes the daily maximum of 2.5-5nm diameter particles, is also not ideal as it shows a considerable number of events just above the 50% ranking value.

We acknowledge that this analysis is pertinent in the article and have included figure E in the supplementary material and added the following text in section 4:

"Finally, the specific range can be adjusted based on site conditions and available instrumentation, it is recommended not to exceed 7 nm to avoid potential non-NPF related disturbances in the signal. A sensitivity analysis of different size range combinations and reduction functions against classical event classification is provided in fig. S1."

We also concede that this metric might necessitate modifications when implemented in diverse locations, as elaborated in our original manuscript and now with a dedicated subsection titled "4 Considerations and limitations of the method and its applicability to other sites" in the newest version.

Furthermore, we briefly discuss the added value of using the active and background regions in the following:

"Furthermore, in urban background environments, ultrafine particle pollution peaks stemming from traffic emissions and those linked to new particle formation (NPF) tend to occur at slightly different times—traffic emissions peak in the morning, while NPF peaks around midday (Trechera et al., 2023). Therefore, a thoughtful selection of both active and background regions, coupled with a reasonable distance of several hundred meters from traffic sources, is likely to enhance the effectiveness of applying this method in these environments."

1.1.4 While discussed in the text, the possibility of such a metric being useful in more complex environments is unlikely due to the multiple sources of particles, which apart from that would also make it somehow more difficult to determine the background regions as well. This should be further discussed as examples of such cases are missing.

Thank you for bringing up an important point. Complex environments such as cities do have multiples sources of sub-5 nm particles such as traffic emissions and other combustion processes. However, the number concentration of these particles, once emitted, has been shown to decay exponentially with distance from the point/line sources. At distances larger than 100 of meters the number concentration of sub-5 nm is very similar to the background (Rönkkö and Timonen, 2019 and references therein). This is not to say that these influences do not have an important effect on the aerosol population and composition. They do modify the background total number concentration and chemical composition; however, due to rapid processes such as dilution, condensation, and dry deposition, primary emissions, or immediately formed particles from exhaust are lost quickly in the smallest sizes. Consequently, their influence on this metric is significantly reduced.

As mentioned in the previous comment, we now have a dedicated section in the manuscript addressing these considerations: 4 Considerations and limitations of the method and its applicability to other sites.

1.1.5 Furthermore, Rönkkö et al., 2017 found that traffic can be a source of 3nm particles. Considering that the expected time of a NPF event can coincide in many cases with increased volumes of traffic, how will traffic derived particles be attributed as non-atmospherically formed in busy urban environments and carry on the analysis.

Please see our answer to the previous comment. However, we have updated the text to mention that "The method will likely not work at roadside or curbside sites as the sub-5nm population is highly sensitive to traffic emissions at these distances (Rönkkö et al., 2017)."

**1.2 Minor comments**

1.2.1 Line 31. NPF events take place with very variable frequency around the sites studied so far. Please rephrase that as the way this is written now it implies that the frequency of the events is similar everywhere.

We have modified line 31 and now it reads: "Atmospheric new particle formation (NPF) events take place, though with variable frequencies, in most of the continental environments …"

1.2.2 Line 244. This needs to be further discussed and examples of ways to filter the factors mentioned should be given.

Thank you for the suggestions. We have now expanded the section with few examples, and it reads as follows:

"Depending on the dataset, one may want to filter out such influencing factors before applying the Nano Ranking Analysis for example by manually removing all days with rain (as done in this study) or blowing snow or by carefully choosing the background region so that the effects of polluted primary sources in the 2.5-5nm range are diminished."

Rönkkö, T., Kuuluvainen, H., Karjalainen, P., Keskinen, J., Hillamo, R., Niemi, J. V., Pirjola, L., Timonen, H. J., Saarikoski, S., Saukko, E., Järvinen, A., Silvennoinen, H., Rostedt, A., Olin, M., Yli-Ojanperä, J., Nousiainen, P., Kousa, A., & Dal Maso, M. (2017). Traffic is a major source of atmospheric nanocluster aerosol. *Porceedings of the National Academy of Sciences*, (29), 7549–7554. https://doi.org/10.1073/pnas.1700830114

**Citation**: https://doi.org/10.5194/ar-2023-5-RC1

**2 RC 2**

The authors present a new method for classifying new particle formation (NPF) based on a simple metric: the number concentration between diamters 2.5 and 5 nm. The new method may be useful when analyzing the occurrence and intensity of NPF events and is definitely more easily automatisized than some of previously presented methods. Prior to being publishable, several issues need to be addressed.

We thank RC2 for their constructive comments and appreciate that the reach of our study is understood.

**2.1 General comments/questions:**

**2.1.1** Why is the ability to predict NPF event frequency important? It is clear that atmospheric models need submodels/parameterizations for NPF rates but it is not explained why event frequency is important and how it could be used?

Thank you for highlighting the need for additional explanation. From a modelling perspective, we agree that NPF rates is the most important parameter. Once the mechanisms of NPF are well understood, NPF rates can be derived from relevant parameters.

From an observational standpoint, traditionally, the frequency of NPF events holds significance because formation rates (J) and growth rates (GR) are typically calculated only when classical events are identified. Therefore, when assessing the importance of NPF from a measurement perspective, it is important to determine both the frequency and intensity of these events (Kerminen et al., 2018).

In our manuscript, we aim to distance ourselves from the traditional "event frequency" concept since Nano Ranking provides a value to describe each day, not a category. However, when categorizing the ranking into modes, we have a concept similar to event frequency. This concept helps us describe a site in terms of the quantity of events relative to their intensity.

Broadly speaking, we need observation data and observation data analysis to understand the mechanisms (precursor, meteorological conditions, etc.) that affect NPF rates. To achieve this, we must have knowledge of the prevailing conditions in which NPF takes place and how many particles it produces. Nano Ranking facilitates achieving this objective using extensive datasets and, consequently, holds the potential to enhance parametrizations.

**2.1.2** The presented method relies only on number concentrations in the interval 2.5 nm - 5 nm, while ealier methods (e.g. Dal Maso et al., 2005) view early growth also as important. This raises the question: what is NPF? How should it be defined? Is it the same as nucleation, or is it apparent nucleation (Kerminen and Kulmala, 2002) at some larger size, which would also imply that the competition between growth and scavenging is also important?

The comment addresses a philosophical aspect of defining NPF. While the broader debate on the precise definition of NPF is indeed pertinent, it is not the central focus

of this paper. We view NPF as encompassing both the clustering process and the growth of these clusters. However, we recognize that interpretations vary: some equate NPF strictly with clustering (or nucleation), while others find it should encompass growth up to certain sizes beyond a few nanometers. The latter interpretation aligns more closely with our understanding of NPF. The chosen 2.5-5 nm size range in our method is particularly sensitive to the growth of clusters, making it a robust indicator of both the occurrence and intensity of NPF. As illustrated in Figure 2, cases with a high-ranking percentile also effectively capture the growth to larger sizes, which traditionally signifies a regional NPF event. For a discussion regarding growth and the comparison of this method to Dal Maso et al. (2005) please see our answer 1.1.1 to RC1.

> 2.1.3 How do you know that this method works also at other sites, especially polluted ones, when it is tested only in Hyytiälä? In the abstract you state that "The new method... is expected to serve as a valuable tool.... at many types of environments" Having only this one site as a test case is, from my point of view, the main weakness of the manuscript. At least, such statements without proper justification, should not be stated. In addition, on lines 54-55 the authors underrate the traditional methods based on them not likely being generally applicable. At the same time the authors state that this new method likely is.. based on analysis of one quite clean finnish site. ("My method is likely better than yours. I have no actual proof, however...")

We appreciate your concerns regarding the general applicability of our method based on data from a single site.

Our confidence in the method's broader applicability comes from preliminary tests we have conducted in diverse environments. Specifically, we have tested the method in mountain top sites such as Chacaltaya, Jungfraujoch, Mount Cimone and Izaña. Additionally, we have applied the method in urban, polluted environments, including Beijing, and El Alto city. In all these scenarios, the results have been promising.

We understand the importance of showcasing the method's versatility across various sites. However, testing the applicability of this method in different environments is out of the scope of this study and we will return to that in future manuscripts which are not yet available for public dissemination as they also include other results.

The reason for presenting the method using only Hyytiälä as a test ground in this manuscript is to lay down the principles of the method in a well characterized station. This understanding will then facilitate its application in other environments, allowing for a more nuanced dissection of the factors influencing NPF.

Accordingly, we have modified the text on lines 54-55 to be more precise on our assessment of other methods and now it reads as follows:

"However, the indicators developed so far tend to be sensitive to the dominant NPF pathway and possibly other site-specific factors, requiring a priori knowledge of the mechanisms involved in NPF at any given site or requiring complementary measurements, such as concentration measurements of sulfuric acid and other precursors gases that are not always available in the field."

Furthermore, we have added the following text to our conclusions to acknowledge that further testing of the method is needed:

"While testing this method in other environments is not yet completed and will be presented in other publications, we envision that the new method is applicable to many other types of environments and, besides providing probabilistic information about the occurrence and intensity of NPF, it has potential to provide valuable insight into the origin of newly formed particles at sites affected by multiple sources of aerosol precursors."

**2.2   More detailed comments:**

**2.2.1   Line 97: How was the time interval 21:00-06:00 chosen? Would this be the window also for other sites?**

Thank you for the observation. The text has now been clarified and reads as follow:

"4. *Find the background number concentration for each day ($N_{B;2.5-5}$)*. The background concentration corresponding to a given day is determined based on the median value of $N_{2.5-5}$ in the so-called background region after applying the 2-hour rolling smoothing of the timeseries (step 2). For Hyytiälä, this time window is between 21:00 and 06:00 (Fig. 1D). The window boundaries are selected so that they contain the seasonal daily median minimums (Fig. 1C). These boundaries are site specific and slightly different values are expected for other sites."

**2.2.2   Section 2.1.1 NPF "mode" fitting: What is the motivation behind these modes, i.e. why not just use the deltaN values as boundaries? What then, if at some other site the distribution would be such that it is not as nicely approximated by a couple of lognormal functions?**

Thank you for your observation. The motivation behind categorizing days into modes rather than using deltaN is to facilitate certain types of analysis where a continuous intensity variable is not optimal, such as comparing days with different synoptic or transport patterns. This categorization provides a clear separation for these purposes.

For instance, consider the Hyytiälä scenario where we have days with low, medium, and high-intensity NPF taking place. Rather than arbitrarily setting thresholds to separate these days, we employ mode fitting to make a data-driven distinction. While it is possible to perform a one-dimensional cluster analysis, we opt for using modes for simplicity.

We acknowledge that the situation at other sites may differ from that at Hyytiälä, but we believe that, in general as it often happens in natural phenomena, a few modes are adequate for a data-driven categorical classification of the site.

Furthermore, having the "modes" facilitates the comparison between sites and we have now added our reasoning in the text (section 2.1):

"The modes also make the comparison between different sites more robust as it is possible to, for example, justify the comparison between "intense events" in different environments (i.e., compare their frequency and intensity) for the highest mode. Such

a comparison would be more challenging using just a specific numerical threshold ($\Delta N_{2.5-5}>$ "certain limit"), as this approach might not account for potential differences in condensation/coagulation sink and other parameters affecting NPF, in addition to NPF itself."

We again thank the reviewer for reminding us that the motivation behind the modes was not properly highlighted in the text.

**2.2.3 Line 152: What does "kernel inversion method" mean? Please explain with a few sentences or give a refence.**

We have now added a reference and the text reads as follows: "The DMPS outputs were inverted following the normal pseudo inversion routine with DMA kernels (Stolzenburg, 1988)"

2.2.5 Equation 3: The notation is again confusing. Usually when there is a sum, there is some increasing index. Also, what is N_Dp2 ? Is it a pointwise size distribution density or the number concentration in some size interval?

We apologize for the lack of clarity in the previous description of equations 2 and 3. In the new version of the manuscript, we have modified the text and equations 2 and 3 to avoid any ambiguity and reads as follows:

"The particle formation rates ($J_{D_{p_i}}$) describes the flux of growing particles past some diameter $D_{p_i}$ and were calculated following the scheme described in Kulmala et al. (2012). The formula is shown below:

$$J_{D_{p_i}} = \frac{dN_{[D_{p_i}, D_{p_u}]}}{dt} + \sum_{l=i}^{u-1} \sum_{j=l}^{max-1} K\left(D_{p_l}, D_{p_j}\right) \cdot N_{[D_{p_j}, D_{p_{j+1}}]} \cdot N_{[D_{p_l}, D_{p_{l+1}}]} + \frac{GR_{[D_{p_i}, D_{p_u}]}}{D_{p_u} - D_{p_i}} \cdot N_{[D_{p_i}, D_{p_u}]}$$

(2)

where the first term stands for observed time derivative of particle number concentration calculated from particle number size distributions (PNSD) from DMPS measurements. In our case we chose $i$ and $u$ such that $D_{p_i} = 3nm$ and $D_{p_u} = 7nm$. The second term describes the coagulational scavenging which considers the sum of coagulation sinks of particles from each size bin where $K$ is the coagulation coefficient of particles in the size $D_{p_l}$ and $D_{p_j}$. The sum is over all bins from the DMPS between $D_{p_i}$ and $D_{p_{max}} = 1000$ nm. A correction factor for particle hygroscopic growth is applied when calculating coagulation sinks as described by Laakso et al. (2004) who developed a hygroscopic growth factor for specifically for Hyytiälä as a function of measured relative humidity. This method can increase the accuracy when estimating coagulation sink of particles over different sizes, where their hygroscopic properties also differ.

The last term in Eq. (2) accounts for particle losses due to growth into larger sizes. We used the maximum concentration method described in Kulmala et al. (2012) to calculate the particle growth rate $GR_{[3,7)}$ from days that were classified as having NPF events (event Ia, Ib and II) based on the guideline illustrated in Dal Maso et al. (2005) which is briefly discussed in the previous section. "

2.2.6 I don't understand the sentence on lines 178-179. Why is a median value for GR used and not a GR for each individual day? And how similar is similar in the previous sentence?

We have now clarified the text in order to make it more understandable and it reads as follows:

"Since determining individual growth rates for non-event days is not possible with the existing methods, we determined the GR for non-event days by calculating a median $GR_{[3,7)}$ based on the method described by Kulmala et al. (2022b). This method first takes the diurnal median PNSD for all non-events and then normalizes each size bin by the maximum value of that bin, thus revealing the hidden growing mode for these non-event days (same normalization is used in Fig. 9). This procedure allows the quantification of the growth rate and subsequently the calculation of $J_3$ for each individual non-event day. In short, for non-event days, we follow the same procedure used to calculate event-day $J_3$ except that instead of using the $GR_{[3,7)}$ for a specific day—which cannot be calculated for single non event days—we use the median $GR_{[3,7)}$ of all non-event days. As comparison, the median $GR_{[3,7)}$ for event and non-event days was 5.1 nm h$^{-1}$ and 3.6 nm h$^{-1}$ respectively."

**2.2.7 Lines 222-223: Instead of "a continuous variable", do you actually mean the rolling median? Or do I understand fig 8 somehow incorrectly?**

Thank you for pointing out the lack of clarity in the text. We mean a continuous variable but have changed the text so that it is clearer now:

"Figure 8 highlights the benefits of utilizing a continuous variable, such as $\Delta N_{2.5-5}$, to characterize new particle formation. This continuous nature enables the application of techniques like a rolling median filter, which is exclusive to non-categorical variables. In the figure, each red point represents a day, while the blue curve results from a 30-day rolling median. This representation distinctly reveals the seasonality of the NPF events, indicating a heightened intensity and/or frequency during spring. Employing a categorical classification for NPF events would complicate this analysis considerably."

**2.2.8 Lines 263-165: Wasn't this recent finding already hypothesized by Kulmala, Pirjola and Mäkelä in their Nature-paper in 2000 (thermodynamically stable clusters)?**

Our understanding is that the cited article "predicted" the presence of clusters which has been confirmed by observations later on. In our thinking NPF (including quiet NPF) refers to cases where clusters have started to grow to larger sizes, either describing local/sub-regional NPF, or forming a regional NPF event if this takes place over large enough spatial scales.

**2.2.9 Lines 269-272: Again the authors claim that this method is likely to be applicable to many other sites also, without proper justification. Analysis with data from two sites of very different types would be much more convincing than Hyytiälä only.**

We understand that given the aim of our article — presenting the new method using only data for one site — we cannot claim that it will work for other sites. However, as mentioned before (2.1.3), we have tested the method in other scenarios with satisfactory results. We have now changed the word "hypothesize" to "envision" in order to make our hypothesis more cautious. Furthermore, we have modified the text in the last sentence of the conclusions to address the lack of method testing in other sites and now it reads:

"While testing this method in other environments is not yet completed and will be presented in other publications, we envision that the new method is applicable to many other types of environments and, besides providing probabilistic information about the occurrence and intensity of NPF, it has potential to provide valuable insight into the origin of newly formed particles at sites affected by multiple sources of aerosol precursors. "

**Citation**: https://doi.org/10.5194/ar-2023-5-RC2

**3 RC3**

The study published by Aliaga et al. concerns the development of a method for automatically providing information on the formation of new particles from observations, both in terms of the occurrence of the process and its intensity. Although its effectiveness is only demonstrated for the boreal forest environment in this study, this method appears to be an interesting alternative/complement to traditional methods that rely on visual detection of events (and are therefore subject to a certain subjectivity); in particular, it overcomes the limitations/difficulties associated with these methods with regard to the identification and characterization of very low-intensity events. I therefore recommend the publication of this work, which I hope will subsequently call for further studies to assess how this approach can be used in environments with contrasting characteristics. The few questions and comments listed below should, however, be considered before publication of this manuscript.

We thank RC3 for the positive remark and appreciate that the reach of our study is understood.

**3.1 General comments/questions**

**3.1.1 P2, L53-55 : « However, the indicators developed **so far** tend to be sensitive to the dominant NPF pathway and possibly other site-specific factors, casting doubts about their general applicability to different atmospheric environments » : I find this wording somewhat disturbing, as it implies that, unlike existing methods, the proposed method is not site-specific... but this is the case, at least as far as the determination of active and background regions is concerned (as clearly mentioned in Sect. 3). Furthermore, there is nothing in this study to demonstrate the applicability of this method (which limitations are actually discussed at the end of Sect. 3) to other environments (e.g. mountain sites affected by complex boundary layer dynamics; urban areas affected by pollution, with background regions possibly changing on a weekend / week day basis).**

Thank you for the relevant observation. We have now changed the text to be more specific on our assessment of other indicators. The text now reads as follows:

"However, the indicators developed so far tend to be sensitive to the dominant NPF pathway and possibly other site-specific factors, requiring a priori knowledge of the mechanisms involved in NPF at any given site or requiring complementary measurements, such as concentration measurements of sulfuric acid and other precursors gases that are not always available in the field."

 P2, L63-64 : About the plan. Since the description of the method is based on the data, I would suggest putting the description of the site and the instrumental set-up before the description of the method. Furthermore, I think Sect. 3 could be split into two sub-sections to improve the clarity of the messages: one associated with the results in the boreal forest, the other dedicated to the limitations of the method / discussion of its use in other environments.

Thank you for the recommendations. We agree that for the sake of clarity it is better to split section 3—Results and Discussions—into two sections that are now also modified in the text. The corresponding titles are:

- 3 Results of applying the method in the boreal forest
- 4 Considerations and limitations of the method and its applicability to other sites

As for the first recommendation, we think that the main aim of the paper is to describe the new method, rather than the fact that we have applied it to Hyytiälä and therefore would like, if possible, to preserve the current structure that in its current form, highlights the method rather than the site.

3.1.3 P2, L68 : About the method in general. For me, a few lines clearly explaining how the method can/should be used are missing. I understand that the information associated with ranking and the determination of the modes first requires a sufficiently long data set, and that the "reference image" resulting from this analysis can then be used to "characterize" new data/individual events, to position them. Is this correct? If so, what are the constraints for defining the reference dataset (length, representativeness of different seasons, etc.)?

Thank you for the observation. It is correct that the method is dependent on the length of the dataset for its relevance. The required duration is indeed contingent upon the specific objectives of the study. For a short campaign, a duration of a few weeks might be sufficient, provided it captures a broad spectrum of potential intensities for the given site and the given objective. On the other hand, when one is analysing trends in NPF intensity and frequency, a dataset spanning multiple years would likely be more appropriate.

Our primary intention with this method is to describe existing datasets rather than to characterize new data points (although the method does hold potential for the latter application).

For clarity we have now included the following paragraph at the end of section 2.1: "We anticipate that these dual outputs will facilitate a deeper understanding of the mechanisms underpinning NPF for specific sites. The former can be compared with continuous parameters linked to NPF, such as precursor gases, condensation sink, meteorological conditions, and the time over land of the associated air mass. In contrast, the latter can be aligned with categorical elements like synoptic patterns, transport mechanisms, volcanic eruptions and intersite comparison. While there is no strict minimum on the number of days required to implement this method, we advise

a baseline of one month to increase the likelihood of obtaining a representative sample of NPF intensity and occurrence. However, more extensive datasets are preferable, with multi-year and multi-season time series being optimal."

> 3.1.4 P5, L146-148: "The CPCs can detect particle number concentrations ranging from 3 nm to 10 nm, whereas the detection of particle size range for the DMPS is 3–1000 nm." The wording does not seem clear to me. I think it should be clear what the cutoff diameter associated with each of the two CPCs is, and then the size range covered by the setup that combines the two CPCs / DMAs.

Thank you for noticing this unclear section. The text has now been modified and reads as follows:

"At the Hyytiälä site, the measurements are taken with a twin-DMPS system consisting of two separate DMPSs operated in parallel. The first DMPS determines the size distribution of small particles, 3–50 nm in diameter, and the other measures larger particles, 15 nm–1000 nm in size. Together they cover the size range of 3–1000 nm. The CPCs used for particle counting are a TSI-3025 and a TSI-3010 (TSI Inc.), respectively. Both DMPSs contain a dryer at the inlet, with a continuous sheath flow with a relative humidity below 30% to enable measuring the particle size under relatively dry conditions. The measurement height of the DMPS system is at ground level at the station (~2 m), and the time resolution of the measurements is 10 minutes. The DMPS outputs were inverted following the normal pseudo inversion routine with DMA kernels described by Stolzenburg (1988). The inverted outputs were used for visualization of the Nano Ranking Analysis percentile bin separation (Fig. 2), as well as for the traditional NPF event classification and calculations of particle formation rates."

> 3.1.5 P5 : About the coagulation sink.
>
> 3.1.5.1 L167-168 : *"The second term describes the particle losses due to coagulation to larger size particles with corresponds to their number concentrations (NDp)."* Check the wording.
>
> 3.1.5.2 Equation 3 : I do not understand the proposed formulation if Dp1 and Dp2 have fixed values; in other words, what does the sum relate to?
>
> 3.1.5.3 L172: *"The correction factor for particle hygroscopic growth is applied when calculating coagulation sinks."* Can the authors say a little more about this correction factor, or at least cite the study describing its calculation?

We apologize for the unclarity of the text. We now have modified the section addressing all 3 points raised by the reviewer and now it reads as follows:

"The particle formation rates ($J_{D_{p_i}}$) describes the flux of growing particles past some diameter $D_{p_i}$ and were calculated following the scheme described in Kulmala et al. (2012). The formula is shown below:

$$J_{D_{p_i}} = \frac{dN_{\left[D_{p_i},D_{p_u}\right)}}{dt} + \sum_{l=i}^{u-1}\sum_{j=l}^{\max-1} K\left(D_{p_l},D_{p_j}\right) \cdot N_{\left[D_{p_j},D_{p_{j+1}}\right)} \cdot N_{\left[D_{p_l},D_{p_{l+1}}\right)} + \frac{GR_{\left[D_{p_i},D_{p_u}\right)}}{D_{p_u} - D_{p_i}}$$
$$\cdot N_{\left[D_{p_i},D_{p_u}\right)}$$

(2)

where the first term stands for observed time derivative of particle number concentration calculated from particle number size distributions (PNSD) from DMPS measurements. In our case we chose $i$ and $u$ such that $D_{p_i} = 3nm$ and $D_{p_u} = 7nm$. The second term describes the coagulational scavenging which considers the sum of coagulation sinks of particles from each size bin where $K$ is the coagulation coefficient of particles in the size $D_{p_l}$ and $D_{p_j}$. The sum is over all bins from the DMPS between $D_{p_i}$ and $D_{p_{max}} = 1000$ nm. A correction factor for particle hygroscopic growth is applied when calculating coagulation sinks as described by Laakso et al. (2004) who developed a hygroscopic growth factor for specifically for Hyytiälä as a function of measured relative humidity. This method can increase the accuracy when estimating coagulation sink of particles over different sizes, where their hygroscopic properties also differ.

The last term in Eq. (2) accounts for particle losses due to growth into larger sizes. We used the maximum concentration method described in Kulmala et al. (2012) to calculate the particle growth rate $GR_{[3,7)}$ from days that were classified as having NPF events (event Ia, Ib and II) based on the guideline illustrated in Dal Maso et al. (2005) which is briefly discussed in the previous section. Since determining individual growth rates for non-event days is not possible with the existing methods, we determined the GR for non-event days by calculating a median $GR_{[3,7)}$ based on the method described by Kulmala et al. (2022b). This method first takes the diurnal median PNSD for all non-events and then normalizes each size bin by the maximum value of that bin, thus revealing the hidden growing mode for these non-event days (same normalization is used in Fig. 9). This procedure allows the quantification of the growth rate and subsequently the calculation of $J_3$ for each individual non-event day. In short, for non-event days, we follow the same procedure used to calculate event-day $J_3$ except that instead of using the $GR_{[3,7)}$ for a specific day—which cannot be calculated for single non event days—we use the median $GR_{[3,7)}$ of all non-event days. As comparison, the median $GR_{[3,7)}$ for event and non-event days was 5.1 nm h[-1] and 3.6 nm h[-1] respectively."

   3.1.6   P6, L183: "*we examine the comparison of ΔN2.5-5 and diurnal patterns*". The variables concerned by the diurnal variations studied should be indicated.

The text has been modified and reads as follow: "Specifically, we examine the comparison of ΔN2.5-5 and diurnal N2.5-5 concentration patterns (Fig. 5), new particle formation rates (Fig. 6), traditional classification of events (Fig. 7), and seasonality (Fig. 8)."

3.1.7   P6, L204 – 206 : "In addition, Fig. 6 demonstrates that the novel Nano Ranking Analysis presented here can be compared with continuous variables, such as the particle formation rate, in investigating NPF." Could the authors elaborate a little more on how the comparison is carried out? How is the value of J included in the comparison obtained: is it a value (mean, median?) calculated over the whole day, over an identified time window?

Thank you for the observation. We have now changed to text specifying how the comparison is made and it reads as follows:

"Previously, studies of continuous variables and their effect on NPF have been constrained by a binary division of days into NPF event and non-event days. Figure 6 demonstrates that the novel Nano Ranking Analysis presented here can be compared with continuous variables, such as the particle formation rate, in investigating NPF. In this case we use the day's maximum $J_3$ and compare it to its corresponding $\Delta N_{2.5-5}$ (fig. 6A) and percentile ranking (fig. 6B). For the location and data set considered here, this daily $J_3$ maximum in general corresponds to a time window similar or very close in time to the time when the max $N_{2.5-5}$ (fig. 1D) is found for each individual day."

**3.2   Minor / technical comments**

Thank you. This is now corrected.

**4   Bibliography**

Dada, L., Chellapermal, R., Buenrostro Mazon, S., Paasonen, P., Lampilahti, J., Manninen, H. E., Junninen, H., Petäjä, T., Kerminen, V.-M., and Kulmala, M.: Refined classification and characterization of atmospheric new-particle formation events using air ions, Atmospheric Chem. Phys., 18, 17883–17893, https://doi.org/10.5194/acp-18-17883-2018, 2018.

Dal Maso, M., Kulmala, M., Riipinen, I., Wagner, R., Hussein, T., Aalto, P. P., and Lehtinen, K.: Formation and growth of fresh atmospheric aerosols: eight years of aerosol size distribution data from SMEAR II, Hyytiälä, Finland, Boreal Environ. Res., 10, 323–336, 2005.

Deng, C., Li, Y., Yan, C., Wu, J., Cai, R., Wang, D., Liu, Y., Kangasluoma, J., Kerminen, V.-M., Kulmala, M., and Jiang, J.: Measurement report: Size distributions of urban aerosols down to 1 nm from long-term measurements, Atmospheric Chem. Phys., 22, 13569–13580, https://doi.org/10.5194/acp-22-13569-2022, 2022.

Ketzel, M. and Berkowicz, R.: Modelling the fate of ultrafine particles from exhaust pipe to rural background: an analysis of time scales for dilution, coagulation and deposition, Atmos. Environ., 38, 2639–2652, https://doi.org/10.1016/j.atmosenv.2004.02.020, 2004.

Kulmala, M., Petäjä, T., Nieminen, T., Sipilä, M., Manninen, H. E., Lehtipalo, K., Dal Maso, M., Aalto, P. P., Junninen, H., Paasonen, P., Riipinen, I., Lehtinen, K. E. J., Laaksonen, A., and Kerminen, V.-M.: Measurement of the nucleation of atmospheric aerosol particles, Nat. Protoc., 7, 1651–1667, https://doi.org/10.1038/nprot.2012.091, 2012.

Rönkkö, T. and Timonen, H.: Overview of Sources and Characteristics of Nanoparticles in Urban Traffic-Influenced Areas, J. Alzheimers Dis., 72, 15–28, https://doi.org/10.3233/JAD-190170, 2019.

**5 Figures**

**5.1 Fig A**

[Figure]

Fig. A. The maximum daily number concentration of particles (y-axis) vs percentile ranking $\Delta N_{2.5-5}$ (x-axis) for different diameter sizes. The midpoint diameter of the plotted size bin is indicated in the title.

**5.2 Fig B**

[Figure]

Figure B. The median of the maximum concentration detected at different particle size bins divided by ranking 5% interval ranking bins as described in fig. 2.

[Figure]

Figure C. Comparison between the Nano Ranking (x axis) and alternative ways of computing the ranking (y axis). Max, Median, and Mean are shown in columns one, two and there, and refer to using the daily maximum, median and mean function, respectively, after applying a two-hour rolling median. The first row applies these functions for the range 2.5-5nm, and the second row applies it to the range 2.5-1000nm.

[Figure]

Figure D. Comparison of the alternative ways of obtaining the ranking against the classical classification from Dal Maso et al. (2005) in the same order described in fig. C.

**5.5 Fig E**

[Figure]

| $D_1[\text{nm}]$ | $D_2[\text{nm}]$ | fun. |
|---|---|---|
| 2.5 | 15 | AB |
| 4 | 7 | AB |
| 3 | 6 | AB |
| 2.5 | 5 | AB |
| 2.5 | 30 | AB |
| 2.5 | 5 | MAX |
| 2.5 | 3 | AB |
| 2.5 | 1000 | MAX |
| 2.5 | 5 | MEAN |
| 2.5 | 1000 | MEAN |
| 2.5 | 1000 | MEDIAN |
| 2.5 | 5 | MEDIAN |

Figure E. Shows the cumulative count of events (Ia, Ib, II; Dal Maso et al., 2005) organized by their ranking values, which are derived from particle number size distribution using various ranking parameters (D1, D2 and fun.). On the y-axis, the cumulative count of these events is displayed, while the x-axis represents the ranking as a percentage. Each curve on the graph corresponds to a different set of parameters used in the ranking. These parameters are detailed in the accompanying table that shows the initial and final diameters used for number concentrations (first and second columns) and the function used to calculate the daily intensity value (third column). These functions include AB, which subtracts the background window average from the active window's maximum value, and MAX, MEAN, and MEDIAN, which use the daily maximum, mean, or median function, respectively. The fourth row (lilac) indicates the recommended parameters for Nano Ranking.